# Cost-Effectiveness of Bivalent Respiratory Syncytial Virus Prefusion F Vaccine for Prevention of Respiratory Syncytial Virus Among Older Adults in Greece

**DOI:** 10.3390/vaccines12111232

**Published:** 2024-10-29

**Authors:** George Gourzoulidis, Charalampos Tzanetakos, Argyro Solakidi, Eleftherios Markatis, Marios Detsis, Diana Mendes, Myrto Barmpouni

**Affiliations:** 1Health Through Evidence, 174 56 Athens, Greece; c.tzanetakos@hte.gr; 2Pfizer Hellas, 154 51 Athens, Greece; argyro.solakidi@pfizer.com (A.S.); eleftherios.markatis@pfizer.com (E.M.); marios.detsis@pfizer.com (M.D.); myrto.barmpouni@pfizer.com (M.B.); 3Pfizer Ltd., Tadworth KT20 7NY, UK; diana.mendes@pfizer.com

**Keywords:** respiratory syncytial virus, vaccination, Greece, RSVpreF

## Abstract

Background/Objectives: To evaluate the health benefits, costs, and cost-effectiveness of vaccination with bivalent respiratory syncytial virus stabilized prefusion F vaccine (RSVpreF) for the prevention of lower respiratory tract disease caused by respiratory syncytial virus (RSV) in Greek adults 60 years of age and older. Methods: A Markov model was adapted to simulate lifetime risk of health and economic outcomes from the public payer’s perspective over a lifetime horizon. Epidemiology, vaccine effectiveness, utilities, and direct medical costs (EUR, 2024) were obtained from published studies, official sources, and local experts. Model outcomes included the number of medically attended RSV cases, stratified by care setting (i.e., hospital, emergency department [ED], outpatient visits [OV]), and attributable RSV-related deaths, costs, life years (LY), quality-adjusted life-years (QALY), and incremental cost-effectiveness ratios (ICERs) of RSVpreF vaccination compared with no vaccination. Results: The model projected 258,170 hospitalizations, 112,248 ED encounters, 1,201,604 OV, and 25,463 deaths related to RSV in Greek older adults resulting in direct medical costs of EUR 1.6 billion over the lifetime horizon. Assuming RSV vaccination would reach the same coverage rates as pneumococcal and influenza programmes, 18,118 hospitalizations, 7874 ED encounters, 48,079 OV, and 1706 deaths could be prevented over the modelled time horizon. The health benefits associated with RSVpreF contributed to an incremental gain of 10,976 LYs and 7230 QALYs compared with no vaccination. The incremental analysis reported that vaccination with RSVpreF was estimated to be a cost-effective strategy resulting in ICERs of EUR 12,991 per LY gained, EUR 19,723 per QALY gained, and EUR 7870 per hospitalized RSV case avoided compared with no vaccination. Conclusions: Vaccination with RSVpreF was a cost-effective strategy for the prevention of RSV disease in Greek adults over 60 years of age. The introduction of RSV vaccination can improve public health by averting RSV cases and deaths and has the potential to fulfil an unmet medical need.

## 1. Introduction

Respiratory syncytial virus (RSV) is a common respiratory virus that usually causes mild, cold-like symptoms (e.g., runny nose, cough, fever, sneezing, and wheezing), but it can also cause serious lower respiratory tract illnesses like bronchiolitis and pneumonia [1,2,3]. RSV is one of the leading causes of respiratory illness in older adults and adults at high risk [4,5,6,7]. There are several risk factors for severe disease in adults infected with RSV, the most important being advanced age and the presence of comorbidities [8,9]. Comorbid conditions that are strongly linked with worse outcomes include cardiopulmonary conditions and immunocompromising conditions [6,8,9,10,11,12].

RSV infection is a global health problem; worldwide in 2015, an estimated 1.5 million episodes of RSV–acute respiratory illness (RSV-ARI) occurred in older adults, of which 14.5% required hospitalization, and 14,000 in-hospital deaths occurred [5]. Despite successful viral clearance, natural immunity against RSV is short-lived and diminishes over time, with reinfection likely to occur throughout life [13,14,15,16,17].

The clinical burden can be severe for adults who are hospitalized with an RSV infection. Among hospitalized adults with RSV infection, severe outcomes occurred in 19.1% of patients, including intensive care unit (ICU) admissions (16.4%), mechanical ventilation (12.4%), and/or death (6.7%) [18]. In a recent multi-country study examining adults ≥ 18 years, 93.6% of overall RSV hospitalizations occurred in adults ≥ 65 years in Greece [19]. Although the study by Osei-Yeboah et al. extrapolated data from other countries to report estimated rates for Greece, these data highlight that older adults are disproportionately affected by RSV-related hospitalizations [19]. Considering the increasing rates of RSV infections, these figures likely represent an underestimation of the current burden in Greece.

Additionally, RSV has a substantial proximal and distal impact on quality of life in older adults and impacts physical functioning, pursuing leisure activities/hobbies, and paid/unpaid work [20]. Apart from the humanistic/clinical burden, RSV infection is a global health issue that places a substantial economic strain on healthcare systems. In 2011–2015, direct healthcare costs of USD 16,034 per episode were attributable to RSV, and the mean length of stay for RSV-associated hospitalization was 7.58 days among US adults 60 years of age and older [21].

Currently, there are no antiviral treatments specific for RSV, and the available treatments are mainly for supportive care [14,16,22]. In August of 2023, the European Commission gave marketing authorization to a bivalent stabilized prefusion F subunit vaccine (RSVpreF) for the prevention of lower respiratory tract disease (LRTD) caused by RSV in adults aged 60 years and older. The efficacy, immunogenicity, and safety of RSVpreF in older adults was evaluated in a placebo-controlled Phase III clinical trial, the “RSV vaccine Efficacy study iN Older adults Immunized against RSV disease” or “RENOIR” trial [23]; however, the potential public health and economic impact of the vaccine in Greece has not been evaluated. Furthermore, economic evaluation plays a vital role in resource-constrained healthcare settings, offering valuable insights into the optimal allocation of scarce resources to maximize public health outcomes. In this light, the objective of the present study was to evaluate the health benefits, costs, and cost-effectiveness of vaccination with RSVpreF for the prevention of LRTD caused by RSV in Greek adults 60 years of age and older.

## 2. Materials and Methods

A decision-analytic model with a Markov structure was locally adapted to simulate the lifetime risk of health and economic outcomes of RSV as well as the expected impact of vaccination against RSV, over a lifetime horizon in a hypothetical population of Greek adults 60 years of age and older. Model outcomes included the number of medically attended RSV cases, categorized by care setting (i.e., hospital, emergency department [ED], outpatient visits [OV]), and attributable RSV-related deaths, costs, life years (LY), quality-adjusted life-years (QALY), and incremental cost-effectiveness ratios (ICERs) of RSVpreF vaccination compared with no vaccination. The perspective of the analysis was that of a Greek public payer (EOPYY) and an annual discounting of 3.5% was applied for future outcomes as often used in such studies in Greece [24,25].

### 2.1. Target Population and Intervention Strategy

The decision-analytic model evaluated a population of Greek adults aged 60 and older, stratified into 5 age groups (60 to 64, 65 to 74, 75 to 84, 75 to 79, and 85 years or older) extracted from the official website of the European Union (Eurostat) and was assumed to be reflective of the calendar year 2023. Within each age group, the population was distributed by comorbidity profile (Table 1), which was defined as persons without chronic or immunocompromising medical conditions (CMC-) and persons with one or more chronic or immunocompromising medical conditions (CMC+) such as chronic cardiac, kidney, liver, and respiratory diseases, diabetes, immunosuppressive diseases, and neurological disorders [26].

Adults were assumed to be either protected against RSV disease through vaccination or to remain unvaccinated and thus unprotected against RSV disease. Overall vaccine coverage was assumed to vary by age and comorbidity profile (Table 1) and was derived from the observational study [27] conducted in Greece, which examined the vaccination coverage of pneumococcal and influenza disease in elderly people ≥ 60 years old.

### 2.2. Model Structure

A population-based Markov model (Figure 1) was locally adapted to represent health and economic outcomes of RSV, as well as the expected impact of vaccination with RSVpreF, over the remaining years of life of a hypothetical population of older adults in Greece. The model population is characterized based on age and comorbidity profile, as described above.

Expected health outcomes are projected for the model population on monthly cycles, from model entry through the end of the modelling horizon, based on age, comorbidity profile, disease/fatality rates, vaccination status, and time since vaccination, accounting for monthly variations in the timing of vaccination and rates of RSV illness. Health outcomes include medically attended RSV, stratified by care setting (i.e., hospital, ED, and OV), and RSV-related deaths. Vaccinated adults are assumed to be at lower risk of future RSV illness. The risk of death from RSV and other (i.e., non-RSV) causes is dependent upon age and comorbidity profile. Indirect effects (i.e., reductions in disease rates due to herd immunity from widespread uptake of RSV vaccine) were not considered.

Expected costs of medical treatment for RSV are generated based on event rates and unit costs in relation to the setting of care (hospital, ED, and OV), age, comorbidity profile, and cost of vaccination.

### 2.3. Model Inputs

The model inputs considered epidemiology (such as rates of RSV disease), mortality, vaccine effectiveness, utilities, and disease-specific direct medical costs for age and comorbidity-specific subgroups. The data, detailed in the following sections, were gathered from published sources and local experts.

#### 2.3.1. Rates of RSV Disease and Mortality

Age- and comorbidity-specific annual rates of RSV disease were considered in the model. More specifically, for respiratory syncytial virus cause of lower respiratory tract infection (RSV-LRTI) requiring hospitalization (RSV-hospitalized), the annual rate was obtained from local experts, while age-specific rates of RSV-LRTI requiring outpatient care (such as ED and OV) were obtained from the literature [28,29] and validated by local experts (Table 1). Rates of RSV were allocated across calendar months based on local experts’ opinions and the distribution by calendar month was assumed to be invariant by care setting (Figure 2).

For mortality, case fatality rate (CFR) due to RSV hospitalization was based on a retrospective study of 30-day mortality in adults aged ≥60 years who tested positive for RSV while in hospital [33]. Since this study reported only a single point estimate for all patients (i.e., CFR by age and/or comorbidity profile was not reported), CFR was allocated across age and comorbidity profile was based on the distribution of mortality by age and risk from a retrospective study of mortality in US adults hospitalized for pneumonia [34]. Moreover, the risk of death due to RSV requiring outpatient care only (i.e., ED or OV) was assumed to be zero.

Age-specific general population mortality rates from WHO Greek life tables [35] were first allocated across comorbidity profiles based on relative risks of mortality (assumed to be 1.75 for CMC+ vs. CMC−); corresponding population weights were also considered in the model.

#### 2.3.2. Vaccine Effectiveness

Vaccination effectiveness (VE) was derived using full season 1 and full season 2 results and post hoc analyses from the RENOIR trial [23]. More specifically, initial VE was assumed to be 84.6% (95% confidence interval (CI): 32.0–98.3) against RSV-Hospital and RSV-ED based on efficacy against medically attended RSV-LRTI with ≥3 symptoms, while efficacy against medically attended acute respiratory illness due to RSV was used as a proxy for VE (65.1% (95% CI: 35.9–82.0)) against RSV-LRTI treated in outpatient settings [23].

Initial effectiveness was assumed to persist (i.e., not to wane) for seven months (average follow-up for full season 1) following vaccine administration. The effectiveness then declined linearly to reach 72.0% (95% CI: 33.4–89.8; RSVpreF: 7 cases, placebo: 25 cases) against RSV-Hospital and RSV-ED and 47.0% (95% CI: 22.7–64.1; RSVpreF: 44 cases, placebo: 83 cases) against RSV-OV in month 16 to reflect the average follow-up of 16.4 months since vaccination for season 2 efficacy [36] (Pfizer Inc., New York, NY, USA, data on file). The rate of linear decline between the end of season 1 and season 2 was assumed to persist through month 41 (end of season 4); thereafter, effectiveness was assumed to be 0% (i.e., from month 42 and beyond). Effectiveness against RSV-Hospital and RSV-ED was truncated at month 42 as a recent RSV vaccine modelling study in older adults projected linear waning of protection that ended at 42 months follow-up post-vaccination [37]. Lastly, the VE was assumed to be invariant by age and comorbidity profile.

#### 2.3.3. Utilities and Disutilities Data

In the absence of local data, age-specific general population utility values were based on EQ-5D VAS scores informed by Janssen and Szende [38]. Disutilities due to RSV were estimated using an area under the curve (AUC) approach and data from two recent studies reporting EQ-5D VAS scores among persons with RSV treated in hospital and outpatient settings, respectively [39,40]. Hence, QALY loss was estimated to be 0.0167 for RSV-Hospital and 0.0054 for RSV-Outpatient. QALY losses were assumed to be invariant by age group and comorbidity profile.

#### 2.3.4. Resource Use and Cost Data

The analysis was conducted from a public payer’s perspective, considering only healthcare costs reimbursed by the public payer. More specifically, the cost inputs considered in the model included vaccination and RSV-attributable direct medical costs estimated for episodes of RSV-Hospital, RSV-ED, and RSV-OV.

To calculate medical costs, except vaccination costs, healthcare resource consumption was combined with unit costs extracted from official sources. Healthcare resource consumption data were gathered from local experts through a questionnaire specifically designed for the study. More specifically, the average resource utilization for Greek patients with RSV was determined based on input from eight local experts. This purpose-built questionnaire was tailored to capture the intricacies of RSV management in Greece, and its content validity was confirmed by the participating experts. The cost model accounted for hospitalizations, emergency department visits, and outpatient care, which included medication use as well as laboratory and diagnostic tests. All costs were expressed to 2024 euros.

The bivalent RSVpreF unit cost per dose was estimated at EUR 205.98 and was obtained from the Greek Ministry of Health [32]. To estimate the age/comorbidity profile related to RSV hospitalization costs, the reimbursement tariff per hospitalization was obtained from the Diagnostic-Related Group (DRG) tariffs’ list issued by the Greek Ministry of Health [30] (Table 1). RSV episodes not included as hospitalization cases were considered outpatient episodes. Therefore, the cost of ED and OV were estimated by combining the resource usage data (such as medications to treat RSV infection, physician visits, laboratory and diagnostic tests) provided by local experts, with the corresponding unit costs extracted from the official website of EOPYY [31] and drug price bulletin [32] (Table 1).

## 3. Model Analyses

Incremental cost-effectiveness ratios are calculated by comparing the vaccination strategy and no vaccination (current standard of care strategy) and calculating the additional cost per additional health benefit in terms of cost per QALY gained, cost per LY gained, and cost per hospitalization avoided.

While Greece does not have an official willingness-to-pay (WTP) threshold for determining the cost-effectiveness of a health intervention, this analysis applied a threshold of EUR 44,000 per outcome. This value is based on published studies suggesting that a health intervention is considered cost-effective if the ICER falls between one and three times the country’s Gross Domestic Product (GDP) per capita [41,42,43], a method commonly used in global health cost-effectiveness studies [24,43,44]. The Greek GDP per capita was estimated at EUR 22,000 based on current prices [45].

To account for statistical uncertainties of multiple key parameters, deterministic sensitivity analyses (DSAs) and probabilistic sensitivity analyses (PSAs) were performed by varying important parameters of the model and sampling various input parameters from appropriate probability distributions.

All major model parameters were tested in a one-way DSA to identify model drivers and examine key areas of uncertainty. The upper and lower bounds tested were calculated as ±25% of the mean value if sensitivity parameters were unavailable. One-way sensitivity analyses were conducted for the following variables: vaccination effectiveness, disutilities, RSV hospital incidence rate, RSV ED incidence rate, and cost inputs. The tornado diagram was used to visually represent the results of sensitivity analysis by identifying the parameters that have the greatest impact on the base case model results. The tornado diagram serves to demonstrate the impact of variations in individual input parameters on the overall model outcomes.

PSAs involve drawing values for each parameter from its individual uncertainty distribution. Contrary to the DSAs, PSAs were performed for all selected parameters simultaneously, with the resulting incremental results recorded. PSAs accounting for uncertainty surrounding key model parameters were used to generate credible intervals (95%) for measures of interest. Typical probability distributions as per the guidance in Briggs et al. [46] were attached to RSV-Hospital incidence rate (Beta), Case-Fatality Rate (Beta), vaccine Effectiveness (Beta), utilities (beta), and direct medical costs (log-normal) for running second-order Monte Carlo simulations in Microsoft excel. An overview of PSA inputs is provided in Appendix A. In total, 1000 simulations were performed, which gave a distribution of incremental results, and consequently, an estimate of the overall uncertainty surrounding the cost-effectiveness results. The results are presented using scatter plots of model simulations on a cost-effectiveness plane (CEP).

To address key uncertainties not covered by the DSAs or PSAs, several specific scenario analyses were performed. Specifically, various age and risk profile groups were considered, including (1) adults aged 60 and older at high risk, (2) adults aged 65 and older at high risk, (3) adults aged 75 and older at high risk, (4) adults aged 60–74 at high risk, (5) all adults aged 65 and older, and (6) all adults aged 75 and older.

## 4. Results

### 4.1. Base Case Model Results

Over the lifetime horizon, the model projected that there would be 258,170 hospitalizations, 112,248 ED encounters, 1,201,604 outpatient visits, and 25,463 deaths related to RSV in Greek adults aged ≥60 years, resulting in direct medical costs of circa EUR 1.6 billion without vaccination. Based on RSV vaccination coverage considered in the analysis, the model indicates that 18,118 hospitalizations, 7874 ED encounters, 48,079 outpatient visits, and 1706 deaths could be prevented over the modelled time horizon (Table 2).

The health benefits associated with RSVpreF vaccination contributed to the incremental gain of 10,976 LYs and 7230 QALYs compared with no vaccination strategy (Table 2). Combining health and economic outcomes, the incremental analysis showed that RSVpreF was estimated to be a cost-effective vaccination strategy that resulted in ICERs of EUR 12,991 per LY gained, EUR 19,723 per QALY gained, and EUR 7870 per RSV hospitalized case avoided compared to no vaccination strategy.

### 4.2. Sensitivity and Scenario Analyses Results

#### 4.2.1. Deterministic Sensitivity Analyses Results

A DSA series was conducted to test how model outputs changed when one parameter was altered at a time. The results of DSA indicated that the base case model results are robust with respect to changes in parameter inputs that are clinically reasonable and are most sensitive to changes in RSVpreF unit cost and RSV rate related to hospitals (Figure 3). It is important to note that in all sensitivity analyses, the RSVpreF vaccine remained a cost-effective strategy, as the ICER per QALY gained stayed below the EUR 44,000 threshold.

#### 4.2.2. Probabilistic Sensitivity Analyses Results

The incremental health outcomes in terms of QALYs gained were plotted against the incremental total costs of the RSVpreF vaccination strategy compared with the no vaccination strategy on the cost-effectiveness plane shown in Figure 4. The PSAs indicated that the total costs of each intervention and QALY yielded were comparable to the base case analyses. The mean ICER on the PSAs was EUR 19,641 for vaccination with RSVpreF compared to no vaccination strategy. Moreover, the PSAs reported that at the threshold of EUR 44,000, vaccination with RSVpreF had a 96% probability of being a cost-effective strategy versus the no vaccination strategy.

#### 4.2.3. Scenario Analyses Results

In all scenario analyses, vaccination with RSVpreF consistently proves to be a cost-effective strategy compared to the no vaccination strategy (Table 3).

## 5. Discussion

Given the significant disease burden associated with RSV, particularly in older adults, there is a clear need for effective prevention strategies. In this light, the current study was conducted from a public-payer perspective to evaluate the potential impact of vaccination with RSVpreF in preventing RSV disease in the older adult population in Greece.

Based on RSV vaccination coverage considered in the analysis, the model indicates that over 74,000 medically attended cases and 1,706 deaths could be prevented over a lifetime horizon of a cohort of adults aged 60 years and over. The effectiveness benefits associated with RSVpreF translate to accruing more QALYs at 22,132,094 QALYs compared with 22,124,864 QALYs with no vaccination strategy. The incremental analysis reported that vaccination with RSVpreF was estimated to be a cost-effective strategy resulting in ICERs of EUR 12,991 per LY gained, EUR 19,723 per QALY gained, and EUR 7870 per hospitalized RSV case avoided compared to no vaccination strategy. All RSVpreF ICERs fell well below the defined cost-effectiveness threshold of EUR 44,000 per QALY gained in Greece versus no vaccination strategy. Moreover, it is important to note that even when applying a more stringent cost-effectiveness threshold of EUR 27,117 per QALY gained [46], as recently proposed in a study conducted in Greece, RSV vaccination would still be considered cost-effective. This indicates that despite the adoption of a more conservative benchmark, vaccination against RSV remains a valuable intervention from an economic standpoint, reinforcing its potential public health and economic benefits.

The sensitivity analyses results indicated that variations in input parameters and assumptions had no significant impact on the base case findings. Specifically, the PSAs estimated a 96% probability that RSVpreF vaccination would be a cost-effective strategy compared to no vaccination, based on a threshold of EUR 44,000 per QALY gained. Additionally, the DSAs demonstrated that across all sensitivity analyses, the RSVpreF vaccine remained cost-effective, as the ICER per QALY gained consistently stayed below the EUR 44,000 threshold.

Moreover, the results of the additional scenario analyses showed that RSVpreF vaccination remained a cost-effective strategy irrespective of the age groups or the comorbidity profiles, which supports vaccination policies that prioritize all older adults. This further underlines the importance of ensuring high vaccination coverage to maximize public health impact.

In August 2024, the RSVpreF vaccine was endorsed by the National Immunization Program for adults, specifically targeting those aged 75 and older, as well as individuals aged 60 and above with risk factors, and therefore is officially reimbursed for these subgroups. This highlights RSVpreF’s significance in meeting the healthcare requirements of these at-risk groups and showcases the acknowledgement by health authorities of RSVpreF’s value [47].

There are only a few studies assessing the impact of RSVpreF vaccination on RSV disease burden, and comparisons with similar studies are limited due to differences in study design, modelling techniques, structural frameworks, underlying assumptions, and data inputs utilized in each study. More specifically, a cost-effectiveness study performed in the USA [48] reported that vaccinating adults 60 years of age and older with RSVpreF could be cost-effective, depending on the price and the durability of vaccine efficacy. Results from this study indicated that achieving a 66% RSV vaccination coverage would substantially alleviate the burden of RSV-related illness. Vaccination of older adults would provide substantial direct health benefits by reducing outcomes associated with RSV-related illness [48]. Moreover, a study conducted in Hong Kong reported that RSVpreF vaccination of adults 60 years of age and older appears to gain QALYs over 2 years, although the cost-effectiveness of RSVpreF is subject to vaccine price and RSV incidence rate [49]. Additionally, the studies [50,51,52], which were conducted in the Netherlands, United Kingdom, and Canada, showed substantial health benefits of RSV vaccination and may well be a cost-effective intervention for older adults.

Despite the differences between studies, all analyses suggest that RSVpreF vaccination of adults aged 60 years and above is likely to substantially contribute to reducing RSV-related illness and can be a cost-effective use of healthcare resources.

At this point, it is important to note that the World Health Organization (WHO) estimates that immunization in general saves 3–5 million lives annually [53], hence it is well established that immunization is the most effective investment in public health, yielding substantial benefits to the whole society [54,55]. Development of an RSV vaccine has been a long-standing public health priority due to the high disease burden imposed by RSV on patients and the healthcare sector [56]. The potential impact of circulating RSV on the healthcare system became particularly evident during the post-COVID-19 era [56]. After protective health measures imposed during the pandemic were gradually lifted, a surge in RSV and other respiratory infections was observed, favoured by increased testing and awareness of respiratory infections, leading to increased pressure on hospitals [57]. Thus, ensuring high vaccination coverage is essential to maximize public health impact, and the inclusion of RSV vaccination in the Greek national immunization program is likely to reach uptake/coverage similar to the pneumococcal or influenza vaccinations among older adults in Greece [27]. Additionally, immunization of the older adult population is of particular importance in Greece, which has one of the highest proportions of older adults worldwide [58]. This makes the Greek population more susceptible to infectious diseases, which can lead to severe health outcomes and increased healthcare costs. The RSVpreF vaccine has the potential to fulfil an unmet medical need; therefore, vaccinating older adults against RSV disease is crucial to protect their health and reduce the overall public health burden.

Given the lack of real-world data on the incidence, resource utilization, and vaccination coverage of RSV in Greece, future research is essential to fill these gaps. Moreover, future research should focus on understanding the barriers to vaccine uptake and coverage among older adults, which would provide insights into improving immunization programs. Exploring the potential integration of RSV vaccination with other vaccines, such as influenza or pneumococcal vaccines, could help increase efficiency and uptake. Moreover, special attention should be given to populations with comorbidities or those in institutionalized settings, as they may be at higher risk for severe outcomes.

In terms of our study limitations, in this analysis, it was assumed that the utility and disutility data obtained from published studies [39,40] were applicable to the healthcare context in Greece. However, due to the absence of locally available data and the limitations in the quality and validity of the relevant studies, this assumption may have constraints. Additionally, local experts were consulted when needed, and their insights, along with any relevant local data, were incorporated to validate the model’s inputs. Despite this, a series of sensitivity analyses confirmed the robustness of the model’s results, as the main findings remained stable across a wide range of parameter variations. Because there was an insufficient number of hospitalized RSV cases in the RENOIR study, the VE against medically attended RSV-LRTI with ≥3 symptoms was used as a surrogate for VE against hospitalized RSV. While there may be concerns of a mismatch, recent early use real-world effectiveness against RSV hospitalization reports a similar magnitude of protection as those reported in the RENOIR clinical trial [36,59,60]. Recent Abrysvo-specific real-world effectiveness (91%) reported by Tartof et al. (Kaiser Permanente) suggests that efficacy data in the RENOIR clinical trial closely mirror its real-world effectiveness in preventing more severe outcomes like hospitalizations [61]. Moreover, the model analysis did not include other potential downstream adverse outcomes and costs associated with RSV (e.g., increased rate of all-cause re-admissions, functional status decline, cardiovascular events, decline in lung function, and transplant-related complications), which likely confers a conservative bias against RSV vaccination. Finally, our model did not consider disutility due to adverse effects of vaccination as severe local or systemic reactions occurred in a very small proportion of the RENOIR participants, with similar proportions between RSVpreF and placebo groups, and were of short durations. Additionally, the present analysis was conducted from the public payer perspective, focusing solely on direct healthcare costs. While adopting a societal perspective could provide a more comprehensive understanding, this analysis did not account for indirect costs such as patient time, caregiver expenses, and productivity losses. These factors represent missed opportunities for society as a whole, suggesting that the true economic burden of RSV may be underestimated, as the analysis only considered healthcare-related expenses.

## 6. Conclusions

To conclude, vaccination with RSVpreF was estimated to be a cost-effective strategy for the prevention of RSV disease in Greek adults aged over 60 years. The availability of RSV vaccination improves public health outcomes by averting RSV cases and deaths and has the potential to fulfil an unmet public health need.

## Figures and Tables

**Figure 1 vaccines-12-01232-f001:**
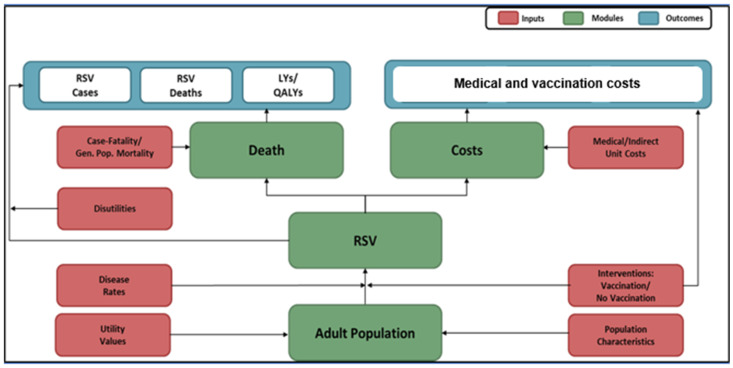
Model schematic.

**Figure 2 vaccines-12-01232-f002:**
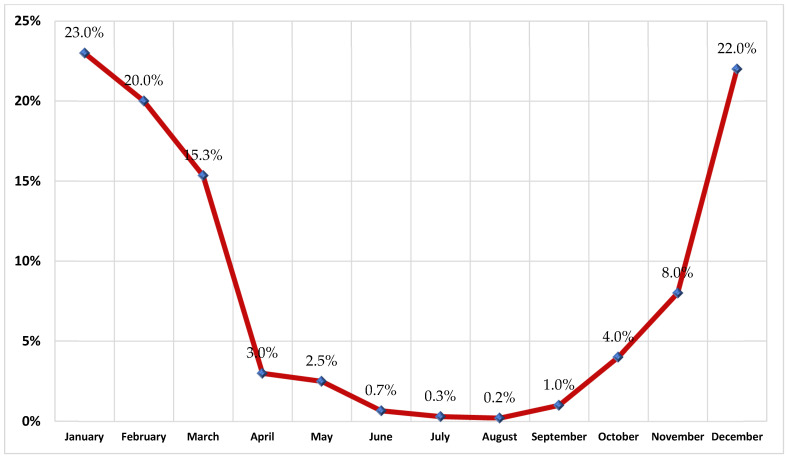
Distribution of RSV cases by calendar month. Source: local clinical experts.

**Figure 3 vaccines-12-01232-f003:**
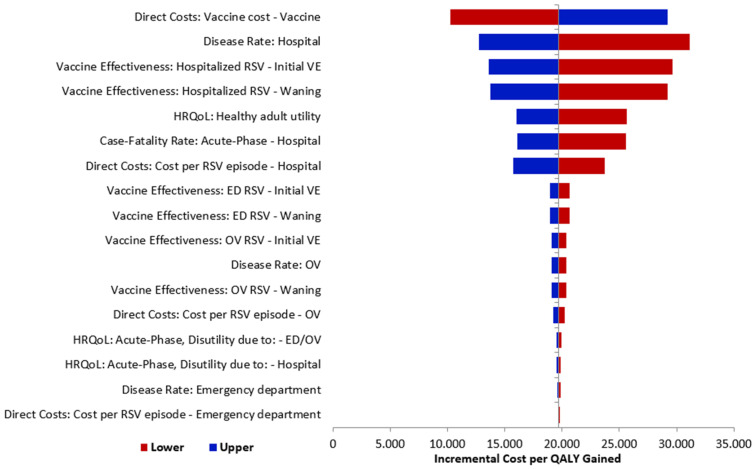
Tornado diagram of RSVpreF vaccination strategy versus no vaccination strategy. RSV: respiratory syncytial virus, OV: outpatient visit, ED: emergency department, VE: vaccine effectiveness.

**Figure 4 vaccines-12-01232-f004:**
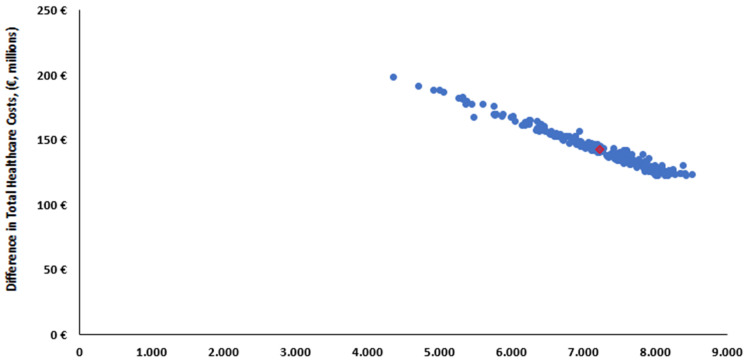
Scatterplot from probabilistic sensitivity analyses results of RSVpreF vaccination strategy versus no vaccination strategy.

**Table 1 vaccines-12-01232-t001:** Model inputs for the comorbidity profile, vaccine coverage, and annual incidence rates of RSV by care setting, age/comorbidity profile, and associated direct medical costs.

Age Group/CP	CP Distribution	Vaccine Coverage by Age and CP	Annual Incidence Rates of RSV (per 100,000) by Care Setting, Age, and CP			RSV-Attributable Direct Medical Costs by Care Setting, Age, and CP		
			Hospital	Emergency Department	OV	Hospital	Emergency Department	OV
60–64								
CMC−	47%	20%	19	80	1496	EUR 1714	EUR 243	EUR 163
CMC+	53%	40%	373	201	2591	EUR 2923	EUR 311	EUR 231
65–74								
CMC−	30%	40%	59	92	1505	EUR 2919	EUR 319	EUR 239
CMC+	70%	50%	558	261	2894	EUR 3881	EUR 388	EUR 308
75–84								
CMC−	16%	50%	150	101	1511	EUR 5111	EUR 319	EUR 239
CMC+	84%	60%	740	310	3143	EUR 7083	EUR 388	EUR 308
85–99								
CMC−	7%	55%	190	111	1517	EUR 7087	EUR 319	EUR 239
CMC+	93%	60%	977	365	3420	EUR 8983	EUR 388	EUR 308

CP: Comorbidity profile, RSV: Respiratory syncytial virus, OV: Outpatient visit, CMC−: persons without chronic or immunocompromising medical conditions, CMC+: persons with one or more chronic or immunocompromising medical conditions. Source: Local clinical experts, Papagiannis et al. [27], McLaughlin et al. [28]., Weycker et al. [29], Diagnosis-Related Group tariffs [30], official website of EOPYY [31], and drug price bulletin [32].

**Table 2 vaccines-12-01232-t002:** Model results of RSVpreF vaccination strategy versus no vaccination strategy.

Parameters	RSVpreF Vaccination Strategy	No Vaccination Strategy	Incremental Results of RSVpreF vs. No Vaccination
Health Outcomes			
No. of cases			
Hospital	240,052	258,170	−18,118
Emergency department	104,374	112,248	−7,874
Outpatient visit	1,153,525	1,201,604	−48,079
Total	1,497,951	1,572,022	−74,071
No. of RSV-related deaths	23,757	25,463	−1,706
Total QALYs	22,132,094	22,124,864	7,230
Total LYs	32,449,292	32,438,315	10,976
Economic Outcomes (in millions)			
Direct cost of vaccine (EUR)	274,22	-	274,22
Direct RSV medical care cost (EUR)	1,421,65	1,553,27	−131,63
Total cost (EUR)	1,695,87	1,553,27	142,59
Cost-effectiveness analysis (RSVpreF vs. No vaccination)			
ICER per QALY gained (EUR)			19,723
ICER per LY gained (EUR)			12,991
ICER per RSV hospitalized case avoided (EUR)			7,870

LY: life years, QALY: quality-adjusted life years, ICER: incremental cost-effectiveness ratio, RSV: respiratory syncytial virus.

**Table 3 vaccines-12-01232-t003:** Scenario analyses results.

Description	RSVpreF Vaccination Versus No Vaccination
(1) Adults aged 60 years and older at high risk	
Incremental costs (in millions, EUR)	EUR 97
Incremental QALYs	6,903
Incremental LYs	10,597
ICER per QALY gained	EUR 14,082
ICER per LY gained	EUR 9,173
(2) Adults aged 65 years and older at high risk	
Incremental costs	EUR 88
Incremental QALYs	6,614
Incremental LYs	10,175
ICER per QALY gained	EUR 13,379
ICER per LY gained	EUR 8,698
(3) Adults aged 75 years and older at high risk	
Incremental costs	EUR 34
Incremental QALYs	4,037
Incremental LYs	6,301
ICER per QALY gained	EUR 8,486
ICER per LY gained	EUR 5,436
(4) Adults aged 60–74 years at high risk	
Incremental costs	EUR 68
Incremental QALYs	3,267
Incremental LYs	4,913
ICER per QALY gained	EUR 20,929
ICER per LY gained	EUR 13,918
(5) All adults aged 65 and older	
Incremental costs	EUR 130
Incremental QALYs	6,930
Incremental LYs	10,541
ICER per QALY gained	EUR 18,795
ICER per LY gained	EUR 12,356
(6) All adults aged 75 and older	
Incremental costs	EUR 49
Incremental QALYs	4,209
Incremental LYs	6,507
ICER per QALY gained	EUR 7,521
ICER per LY gained	EUR 11,625

## Data Availability

All input data for the study are available in the tables published in this manuscript.

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
