# Peer review of "Cost-Effectiveness of Bivalent Respiratory Syncytial Virus Prefusion F Vaccine for Prevention of Respiratory Syncytial Virus Among Older Adults in Greece"

_vaccines, 2024, doi:10.3390/vaccines12111232_

Round 1

Reviewer 1 Report

Comments and Suggestions for Authors

The article presents the results and discussion of a model-based cost-effectiveness analysis of vaccination with the RSVpreF vaccine in older adult in Greece.  The study found an incremental cost-effectiveness ratio (ICER) of €12,991 per LY gained, €19,723 per QALY gained compared to no vaccination. The Method/rationale used

The main problem is that Section 2.3.2 “Vaccine effectiveness” has a critical mistake concerning the method/rationale used in the study for estimating the VE against hospitalization, and there are doubts about the method/rationale used for estimating the VE against emergency department (ED) and outpatient visits (OV) using data obtained in the RENOIR trail.

The critical mistake concerning the VE against hospitalization makes it necessary to do the analysis again using the VE in preventing hospitalization obtained recently instead of the VE assumed in the study.

Critical weaknesses of the study and article:

1. Section 2.3.2 includes a critical mistake concerning the values of vaccine effectiveness (VE) assumed in the study against hospitalization.

Using the VE against medically-attended RSV-LRTI with ≥3 symptoms in the RENOIR trail as an estimate of the VE against RSV-Hospital and RSV-ED in the CE study is a critical mistake because the outcome “medically-attended RSV-LRTI” is not equivalent to hospitalization. The RENOIR trail was developed to assess the VE against RSV-LRTI cases and severe RSV-LRTI (hospitalization due to RSV-LRTI or new/increased oxygen supplementation or mechanical ventilation). RSV-LRTI cases were defined by ≥3 LRTI signs/symptoms. The outcome “medically attended RSV-associated LRTD” was defined as LRTD prompting “any outpatient or inpatient visit such as hospitalization, emergency department visit, urgent care visit, home health care services, primary care physician office visit, pulmonologist office visit, specialist office visit, other visit, or telehealth contact.” The problem is that you cannot use the VE in preventing a RSV case or in preventing a medically-attended RSV-LRTI case in the RENOIR trial as the VE in preventing hospitalization. In fact, the RENOIR study was not powered to estimate efficacy against hospitalization (intervention group = one event; control group = three events) or death (no events) (MWR Morb Mortal Wkly Rep 2023;72:793-801).

A recent study (JAMA. Published online September 4, 2024. doi:10.1001/jama.2024.15775) found a VE for RSV vaccine in preventing hospitalization among individuals aged 60 years and over of 75% (95% CI: 50-87%). This VE against hospitalization is a consistent one, instead of the VE incorrectly assumed in the CE study of 84.6%, 95% CI: 32.0-98.3). 

Consequently, the CE analysis must be done again using the consistent value of VE against hospitalization obtained in the study published in JAMA or in other studies.

Other problems:

2. The title does not explain clearly the content of the article. The study objective was not to assess the value of RSC vaccination but the cost-effectiveness of SRV vaccination in preventing respiratory syncytial virus disease in older adult in Greece. Therefore, the title must be “Cost-effectiveness of respiratory syncytial virus vaccination in older adults in Greece” or a similar one to avoid confusion.   

3. Abstract. In the last sentence of the Results the terms “and €7,870 per RSV hospitalized case avoided” must be deleted. This sentence is about the CE, and this data is not real-world data but data derived from the assumptions about incidence and vaccine effectiveness in preventing hospitalization.

4. It is mentioned that the vaccination effectiveness (VE) was derived using full season 1 and full season 2 results and post hoc analyses from the RENOIR trial. What does it mean?

5. It is necessary to explain in Section 2.3.2 why the VE against “medically-attended RSV-LRTI” in the RENOIR study can be the VE against emergency department (ED) visits due to RSV infections. The outcome “medically attended RSV-associated LRTD” in in then RENOIR trial was defined as LRTD prompting “any outpatient or inpatient visit such as hospitalization, emergency department visit, urgent care visit, home health care services, primary care physician office visit, pulmonologist office visit, specialist office visit, other visit, or telehealth contact.”

6. It is necessary to explain in Section 2.3.2 why the VE against medically attended acute respiratory illness (ARI) due to RSV in the RENOIR study can be a proxy of the initial VE against RSV-LRTI treated in outpatient settings (65.1%, 95% CI: 35.9-82.0).

7. Lines 110-124. The explanation about the model structure can be improved by including a figure with the model structure.

8. The Discussion section does not include an explanation about the study limitations and how they were solved. This explanation is necessary because if the VEs for hospitalization, ED and OV are incorrect or biased, the CE analysis will be incorrect.

9. Line 233-230. Probability sensitivity analysis. The methods used to develop this analysis are not explained in sufficient detail. The methods of a PSA must include a table with the information about the distribution type and parameters for the study variables included on the PSA. Was a Montecarlo simulation used in the PSA? Was the PSA developed using the Excel program?  

10. Line 259. Mention of Figure 2 seems incorrect. 

Comments on the Quality of English Language

The English is correct. 

Author Response

We would like to thank you for your recent correspondence indicating that you are willing to reassess our manuscript entitled: “Cost-Effectiveness of Bivalent Respiratory Syncytial Virus Prefusion F Vaccine for Prevention of Respiratory Syncytial Virus Among Older Adults in Greece.”

We submit the revised manuscript and a point-by-point response to reviewers’ comments (see below). We hope that you will find our responses and modifications to be appropriate. Thank you for your willingness to reassess our revised manuscript.

Reviewer: 1

Reviewer(s)' Comments to Author:

The article presents the results and discussion of a model-based cost-effectiveness analysis of vaccination with the RSVpreF vaccine in older adult in Greece.  The study found an incremental cost-effectiveness ratio (ICER) of €12,991 per LY gained, €19,723 per QALY gained compared to no vaccination. The Method/rationale used

The main problem is that Section 2.3.2 “Vaccine effectiveness” has a critical mistake concerning the method/rationale used in the study for estimating the VE against hospitalization, and there are doubts about the method/rationale used for estimating the VE against emergency department (ED) and outpatient visits (OV) using data obtained in the RENOIR trail.

The critical mistake concerning the VE against hospitalization makes it necessary to do the analysis again using the VE in preventing hospitalization obtained recently instead of the VE assumed in the study.

Critical weaknesses of the study and article:

  1. Section 2.3.2 includes a critical mistake concerning the values of vaccine effectiveness (VE) assumed in the study against hospitalization.

Using the VE against medically-attended RSV-LRTI with ≥3 symptoms in the RENOIR trail as an estimate of the VE against RSV-Hospital and RSV-ED in the CE study is a critical mistake because the outcome “medically-attended RSV-LRTI” is not equivalent to hospitalization. The RENOIR trail was developed to assess the VE against RSV-LRTI cases and severe RSV-LRTI (hospitalization due to RSV-LRTI or new/increased oxygen supplementation or mechanical ventilation). RSV-LRTI cases were defined by ≥3 LRTI signs/symptoms. The outcome “medically attended RSV-associated LRTD” was defined as LRTD prompting “any outpatient or inpatient visit such as hospitalization, emergency department visit, urgent care visit, home health care services, primary care physician office visit, pulmonologist office visit, specialist office visit, other visit, or telehealth contact.” The problem is that you cannot use the VE in preventing a RSV case or in preventing a medically-attended RSV-LRTI case in the RENOIR trial as the VE in preventing hospitalization. In fact, the RENOIR study was not powered to estimate efficacy against hospitalization (intervention group = one event; control group = three events) or death (no events) (MWR Morb Mortal Wkly Rep 2023;72:793-801).

A recent study (JAMA. Published online September 4, 2024. doi:10.1001/jama.2024.15775) found a VE for RSV vaccine in preventing hospitalization among individuals aged 60 years and over of 75% (95% CI: 50-87%). This VE against hospitalization is a consistent one, instead of the VE incorrectly assumed in the CE study of 84.6%, 95% CI: 32.0-98.3).

Consequently, the CE analysis must be done again using the consistent value of VE against hospitalization obtained in the study published in JAMA or in other studies.

Reply: Thank you for your insightful feedback.  You raise a valid point regarding the differences between the outcomes assessed in the RENOIR trial, specifically between medically-attended RSV-LRTI cases and hospitalization. RENOIR trial primary endpoint was to assess vaccine efficacy (VE) against RSV-LRTI with at least 2 or at least 3 signs or symptoms.  Vaccine efficacy against severe RSV-LRTIs like those requiring hospitalization or oxygen supplementation was a secondary endpoint, therefore not powered to estimate VE specifically for hospitalization alone. However, recent real-world effectiveness on hospitalization[1] is now emerging, and despite its limitations (severe comorbid population with 24% immunocompromised and median of 5 for Charlson Comoribidity Index) and the short follow-up period (median time from vaccination to illness of 84 days), the real-world effectiveness data are largely aligned with the RSV-LRTI VE observed in the RENOIR trial for medically-attended RSV-LRTI cases with ≥3 symptoms. 

The Kaiser Permanente study[2] demonstrated that the effectiveness of Abrysvo against RSV-related hospitalization or emergency department (ED) visits was 89% (95% CI, 52%-97%). Moreover, the vaccine effectiveness of ABRYSVO against RSV-associated hospitalizations, ED visits, and critical illness among adults 60 years of age and older was evaluated in an observational study of electronic health records (EHRs) in the Virtual SARS-CoV-2, Influenza, and Other Respiratory Viruses Network (VISION). Among immunocompetent adults 60 years of age and older, vaccine effectiveness of ABRYSVO was 80% (95% CI, 69%-89%) against RSV-associated hospitalizations (n = 28,270)[3]. In addition, the vaccine effectiveness of ABRYSVO against documented RSV infection among adults 60 years of age and older was evaluated in an observational study of EHR data from all Veterans Health Administration (VHA) facilities (N = 101,542)[3]. Vaccine effectiveness of ABRYSVO was 82% (95% CI, 69%-89%) against RSV-associated hospitalization. And the most recently study, the vaccine effectiveness of RSV vaccines (i.e., ABRYSVO or AREXVY) against RSV-associated hospitalizations among adults 60 years of age and older (N = 2,978) was evaluated in an observational study of EHR data from the IVY network. The vaccine effectiveness of RSV vaccination was 75% (95% CI, 50%-87%) against RSV-associated hospitalizations.

These findings are in line with the RENOIR trial’s VE for RSV-LRTI cases with ≥3 symptoms, suggesting consistency between clinical trial efficacy and real-world effectiveness. Moreover, in Surie et al.'s presentation to the CDC ACIP in June 2024 (same author with the reviewer reported study of Jama journal), specifically pointed out the alignment between RCT-derived efficacy data, such as that from the RENOIR trial, and the emerging real-world effectiveness against hospitalization[3].

Given that RCTs are the gold standard to assess the efficacy and safety of new interventions and this alignment between RWE and RCT data, it would be reasonable to maintain the current VE estimate based on RSV-LRTI cases with ≥3 symptoms from the RENOIR trial. We also choose to use the clinical trial data because we have trial follow-up data that is much longer than the short follow-up (2-4 months since vaccination) reported in the real-world studies. 

The RENOIR trial data has been utilized in previous cost-effectiveness studies as a proxy for estimating vaccine efficacy against more severe outcomes, such as hospitalization [4, 5]. Hence, considered appropriate for these analyses which informed the US Advisory Committee on Immunization Practices (ACIP). In conclusion, while the RENOIR trial may not have been powered to directly measure VE against hospitalization, its data can still be appropriately used in cost-effectiveness analysis, in line with established practices and early real world effectiveness studies support similar protection against hospitalization.We expanded the limitations section of the manuscript discussion on this matter.

Other problems:

  1. The title does not explain clearly the content of the article. The study objective was not to assess the value of RSC vaccination but the cost-effectiveness of SRV vaccination in preventing respiratory syncytial virus disease in older adult in Greece. Therefore, the title must be “Cost-effectiveness of respiratory syncytial virus vaccination in older adults in Greece” or a similar one to avoid confusion.

Reply: Thank you for your suggestion regarding the title of the study. The title has been changed.

  1. Abstract. In the last sentence of the Results the terms “and €7,870 per RSV hospitalized case avoided” must be deleted. This sentence is about the CE, and this data is not real-world data, but data derived from the assumptions about incidence and vaccine effectiveness in preventing hospitalization.

Reply: Thank you for your feedback. We would like to clarify the reasoning behind the inclusion of the statement regarding the cost per RSV hospitalization avoided. While we understand the concern that this figure is based on assumptions about incidence and vaccine effectiveness, this is standard practice in cost-effectiveness analysis (CEA). CEAs often rely on modeled data derived from these types of assumptions in order to evaluate potential outcomes, particularly in situations where real-world data is limited or unavailable. The ICER of €7,870 per RSV hospitalized case avoided is meant to provide a clear understanding of the estimated economic impact of preventing a hospitalization through vaccination, which is central to the theoretical framework of cost-effectiveness analysis. The figure does not claim to represent real-world data but rather to provide insight into the projected incremental cost per hospitalization prevented by the intervention under study. This type of projection is widely accepted in CEAs as it allows decision-makers to weigh the potential costs and benefits of public health interventions even before extensive real-world data is available. Therefore, we believe that this statement remains valid and important for interpreting the cost-effectiveness results.

  1. It is mentioned that the vaccination effectiveness (VE) was derived using full season 1 and full season 2 results and post hoc analyses from the RENOIR trial. What does it mean?

Reply: The RENOIR trial followed participants through two RSV seasons (seasonal periods when RSV infections typically occur). The VE was derived from the results over the entire duration of these two seasons, which helps capture the vaccine’s efficacy over a long-term period rather than just a partial season. By using full season data, the study aims to provide a more comprehensive and accurate picture of how well the vaccine works in preventing RSV disease throughout the entire period of exposure to RSV. Additionally, post hoc analyses refer to additional analyses conducted after the trial data have been collected. These analyses are not pre-specified in the study design but are performed later to explore specific questions or patterns that may not have been the primary focus of the original analysis. In this case, the post hoc analyses likely involved a closer look at the VE in specific subgroups of patients or across different outcomes (e.g., hospitalization, medically-attended RSV, etc.) to refine the understanding of how effective the vaccine is in various contexts.

  1. It is necessary to explain in Section 2.3.2 why the VE against “medically-attended RSV-LRTI” in the RENOIR study can be the VE against emergency department (ED) visits due to RSV infections. The outcome “medically attended RSV-associated LRTD” in then RENOIR trial was defined as LRTD prompting “any outpatient or inpatient visit such as hospitalization, emergency department visit, urgent care visit, home health care services, primary care physician office visit, pulmonologist office visit, specialist office visit, other visit, or telehealth contact.”

Reply: Thank you for your comment. The use of the vaccine efficacy (VE) against "medically-attended RSV-LRTI" from the RENOIR study as a proxy for VE against emergency department (ED) visits is based on the definition of "medically-attended RSV-associated LRTI" used in the RENOIR trial. In the trial, this outcome included any outpatient or inpatient visit prompted by RSV lower respiratory tract infection (LRTI), which encompassed a broad range of healthcare encounters, such as hospitalizations, emergency department visits, urgent care visits, and even telehealth contacts. Since ED visits are explicitly included in the definition of "medically-attended RSV-associated LRTI," the VE reported in the RENOIR trial inherently captures the vaccine’s efficacy in preventing not only milder cases that might require outpatient visits but also more severe cases that lead to emergency department visits. While the study was not specifically powered to isolate ED visits as a separate outcome, the inclusion of ED visits within the broader category of medically-attended RSV infections makes it reasonable to apply the VE against "medically-attended RSV-LRTI" as an estimate for VE against ED visits due to RSV.

  1. It is necessary to explain in Section 2.3.2 why the VE against medically attended acute respiratory illness (ARI) due to RSV in the RENOIR study can be a proxy of the initial VE against RSV-LRTI treated in outpatient settings (65.1%, 95% CI: 35.9-82.0).

Reply: Thank you for your comment. The use of the vaccine efficacy (VE) against medically attended acute respiratory illness (ARI) due to RSV in the RENOIR study as a proxy for the VE against RSV-LRTI treated in outpatient settings is supported by the overlap between these two clinical outcomes. In the RENOIR trial, medically attended ARI due to RSV refers to respiratory illnesses caused by RSV that require medical attention, regardless of severity. This broad category includes both lower respiratory tract infections (LRTI) and upper respiratory tract infections (URTI). Therefore, the VE against ARI captures the vaccine's efficacy in preventing both mild and moderate to severe RSV infections that prompt medical intervention, including those treated in outpatient settings. The specific VE of 65.1% (95% CI: 35.9-82.0) for RSV-LRTI treated in outpatient settings is a subset of medically attended ARI, representing cases where the infection progressed to LRTI and required outpatient treatment. Since medically attended ARI encompasses a range of severities, including LRTI treated in outpatient settings, the VE against ARI can serve as a reliable proxy for the VE against outpatient-treated RSV-LRTI. While the outcomes are slightly different, both reflect the vaccine's overall capacity to reduce RSV-related illnesses requiring medical care. This approach is consistent with practices in vaccine studies where more granular data for specific outcomes (like outpatient LRTI) are part of broader categories (like medically attended ARI).Moreover, same data were used in previous cost-effectiveness studies [4, 5].

  1. Lines 110-124. The explanation about the model structure can be improved by including a figure with the model structure.

Reply: Thank you very much for the comment. The figure was added to the manuscript line 135.

  1. The Discussion section does not include an explanation about the study limitations and how they were solved. This explanation is necessary because if the VEs for hospitalization, ED and OV are incorrect or biased, the CE analysis will be incorrect.

Reply: Thank you. Study limitations were expanded to address this comment in the manuscript Discussion section (lines 385-410).

  1. Line 233-230. Probability sensitivity analysis. The methods used to develop this analysis are not explained in sufficient detail. The methods of a PSA must include a table with the information about the distribution type and parameters for the study variables included on the PSA. Was a Montecarlo simulation used in the PSA? Was the PSA developed using the Excel program?

Reply: The PSA was developed using Monte Carlo simulation in Excel. The Methods section of the manuscript was expanded on this matter (lines 245-252), and a table with the distribution types and parameters for each variable was included in Electronic Supplementary Material. Thank you for bringing this to our attention.

  1. Line 259. Mention of Figure 2 seems incorrect.

Reply: Thank you for pointing that out. The figure number has been corrected.

References

  1. Surie, D., W. H. Self, Y. Zhu, K. A. Yuengling, C. A. Johnson, C. G. Grijalva, and F. S. Dawood. "Rsv Vaccine Effectiveness against Hospitalization among Us Adults 60 Years and Older." Jama 332, no. 13 (2024): 1105-07.
  2. Iona Munjal. "Rsvpref Adult Clinical Development Update." https://www.cdc.gov/acip/downloads/slides-2024-06-26-28/02-RSV-Adult-Munjal-508.pdf (
  3. Surie, Diya. "Effectiveness of Adult Respiratory Syncytial Virus Vaccines, 2023–2024. Centers for Disease Control and Prevention." https://www.cdc.gov/vaccines/acip/meetings/downloads/slides-2024-06-26-28/07-RSV-Adult-Surie-508.pdf (
  4. Moghadas, S. M., A. Shoukat, C. E. Bawden, J. M. Langley, B. H. Singer, M. C. Fitzpatrick, and A. P. Galvani. "Cost-Effectiveness of Prefusion F Protein-Based Vaccines against Respiratory Syncytial Virus Disease for Older Adults in the United States." medRxiv (2023).
  5. Hutton, D. W., L. A. Prosser, A. M. Rose, K. Mercon, I. R. Ortega-Sanchez, A. J. Leidner, F. P. Havers, M. M. Prill, M. Whitaker, L. E. Roper, J. Pike, A. Britton, and M. Melgar. "Cost-Effectiveness of Vaccinating Adults Aged 60 years and Older against Respiratory Syncytial Virus." Vaccine 42, no. 24 (2024): 126294.

Reviewer 2 Report

Comments and Suggestions for Authors

I would like to congratulate the authors for developing this novel research, which undoubtedly add knowledge to the economic analyses of vaccines. The research aims to evaluate the health benefits, costs and cost-effectiveness of vaccination with bivalent respiratory syncytial virus stabilized prefusion F vaccine (RSVpreF), for the prevention of lower respiratory tract disease caused by respiratory syncytial virus (RSV) in Greek adults 60 years of age and older.

The authors have comprehended and performed the research based on available literature /information and expert opinion. The methodology has been impeccably designed and the procedures followed in the estimation of costs, cost-effectiveness and health benefits has been presented adequately.

The results of the data analysis have been found adequate, and the way inferences drawn from the analysis is commendable. The tables and diagrams are used for presenting the results. Discussion section has been presented adequately and corroborated with findings of other studied in this field. Conclusion has been drawn from the study results and discussion. References used are found to be adequate.

However, the following comments need to be considered by the authors.

1.      In introduction section, the importance of conducting this economic study (costs, cost-effectiveness, and health benefits) should also be highlighted. (Lines 66-76)

2.     Section 2.3.4. Resource use and cost data – more details on direct medical costs calculation required. ie. disaggregated information of direct medical costs

3.     In methodology it is mentioned that the analysis was conducted from a public payer's perspective, considering only healthcare costs reimbursed by the public payer. Please provide the justifications

4.     In methodology -  To calculate medical costs except for vaccination cost, healthcare resource consumption was combined with unit costs extracted from official sources- more details on resource consumption be added.

5.      Healthcare resource consumption data was gathered from local experts through a questionnaire – brief details rewired

6.     Figure 1. Distribution of RSV cases by calendar month and another Figure 1. Tornado diagram of RSVpreF vaccination strategy versus no vaccination strategy. Please check figure numbers. Also, the purpose of using Tornado diagram needs to be mentioned in the text.

7.     Any limitations in following certain assumptions in this study?

8.     What are the policy implications /recommendations to the government?

9.     Any recommendations for future research ?

Author Response

We would like to thank you for your recent correspondence indicating that you are willing to reassess our manuscript entitled: “Cost-Effectiveness of Bivalent Respiratory Syncytial Virus Prefusion F Vaccine for Prevention of Respiratory Syncytial Virus Among Older Adults in Greece.”

We submit the revised manuscript and a point-by-point response to reviewers’ comments (see below). We hope that you will find our responses and modifications to be appropriate. Thank you for your willingness to reassess our revised manuscript.

Reviewer: 2

I would like to congratulate the authors for developing this novel research, which undoubtedly add knowledge to the economic analyses of vaccines. The research aims to evaluate the health benefits, costs and cost-effectiveness of vaccination with bivalent respiratory syncytial virus stabilized prefusion F vaccine (RSVpreF), for the prevention of lower respiratory tract disease caused by respiratory syncytial virus (RSV) in Greek adults 60 years of age and older.

The authors have comprehended and performed the research based on available literature /information and expert opinion. The methodology has been impeccably designed and the procedures followed in the estimation of costs, cost-effectiveness and health benefits has been presented adequately.

The results of the data analysis have been found adequate, and the way inferences drawn from the analysis is commendable. The tables and diagrams are used for presenting the results. Discussion section has been presented adequately and corroborated with findings of other studied in this field. Conclusion has been drawn from the study results and discussion. References used are found to be adequate.

Reply: We would like to thank the reviewer for the kind words. We truly appreciate it!

However, the following comments need to be considered by the authors.

  1. In introduction section, the importance of conducting this economic study (costs, cost-effectiveness, and health benefits) should also be highlighted. (Lines 66-76).

Reply: Thank you for your feedback. In introduction section in lines 66-76 more information was added to better highlight these key aspects.

2.Section 2.3.4. Resource use and cost data – more details on direct medical costs calculation required. ie. disaggregated information of direct medical costs.

Reply: Thank you for your comment. We have provided more details on direct medical costs expanding this section on lines 203-218

3.In methodology it is mentioned that the analysis was conducted from a public payer's perspective, considering only healthcare costs reimbursed by the public payer. Please provide the justifications.

Reply: Conducting the analysis from a public payer's perspective ensures that the study provides relevant and actionable insights for healthcare policymakers, aligns with the structure of the national healthcare system, and addresses the budgetary considerations essential for the sustainability of public health interventions. This is the acceptable perspective based on the Greek HTA committee, hence similar published Greek studies have adopted the same perspective[1].

4.In methodology -  To calculate medical costs except for vaccination cost, healthcare resource consumption was combined with unit costs extracted from official sources- more details on resource consumption be added.

Reply: Thank you for your comment. Τhe following detail was added in lines 203-209

5.Healthcare resource consumption data was gathered from local experts through a questionnaire – brief details rewired.

Reply: Thank you for your comment. Τhe following detail was added in lines 215-2018

6.Figure 1. Distribution of RSV cases by calendar month and another Figure 1. Tornado diagram of RSVpreF vaccination strategy versus no vaccination strategy. Please check figure numbers. Also, the purpose of using Tornado diagram needs to be mentioned in the text.

Reply: Thank you for pointing that out. The figure number has been corrected. The purpose of using tornado diagram was added in section 3 “model analyses”.

7.Any limitations in following certain assumptions in this study?

Reply: Additional limitations were addressed in the discussion section, specifically in lines 381–422.

8.What are the policy implications /recommendations to the government?

Reply: In the Discussion section specifically in lines 376–402 we reported our policy implications/ recommendations

9.Any recommendations for future research ?

Reply: Thank you for raising this. Recommendations for future research have been added to the Discussion section in the manuscript lines 391-399

Reference

  1. Tzanetakos, C., and G. Gourzoulidis. "Does a Standard Cost-Effectiveness Threshold Exist? The Case of Greece." Value Health Reg Issues 36 (2023): 18-26.

Round 2

Reviewer 1 Report

Comments and Suggestions for Authors

Your responses to the two critical questions are not adequate. The following two questions require a consistent response: 1) Why the VE against “any outpatient or inpatient visit” in the RENOIR trial is equivalent to the VE against hospitalization and the VE against ED visit, if the RENOIR was not able to assess the VE against these outcomes? 2) Why the VE against hospitalization obtained in the JAMA study cannot be a better and consistent assumption for the VE in your CE study? As your responses to these two questions are not clear and consistent, the CE analysis can be potentially incorrect or biased.  

Consequently, the CE analysis must be done again using consistent values of VE against hospitalization and ED visits (and OV visits) obtained in the Suri review, JAMA study or in other studies.

You comment that “Vaccine efficacy against severe RSV-LRTIs like those requiring hospitalization or oxygen supplementation was a secondary endpoint, therefore not powered to estimate VE specifically for hospitalization alone. However, recent real-world effectiveness on hospitalization [1] is now emerging, and despite its limitations (severe comorbid population with 24% immunocompromised and median of 5 for Charlson Comoribidity Index) and the short follow-up period (median time from vaccination to illness of 84 days), the real-world effectiveness data are largely aligned with the RSV-LRTI VE observed in the RENOIR trial for medically-attended RSV-LRTI cases with ≥3 symptoms.”  supports my critical comment. The reference (1) in your response is the JAMA study where the VE against hospitalization was 75% (95% CI: 50-87%). In fact, the JAMA study shows that real-world data for VE against hospitalization can be different than the VE against medically-attended RSV-LRTI cases with ≥3 symptoms in the RENOIR trial.  The VE of 75% (95% CI: 50-87%) is 12.8% lower than the VE of 84.6% (95% CI: 32.0-98.3) in your CE study. Again, VE against “any outpatient or inpatient visit” in the RENOIR trial cannot be equivalent to the VE against hospitalization, unless this equivalence is demonstrated in a consistent way. This demonstartion has nor been presented in the Methods section. 

The Surie review (reference 3 in your response) supports my critical comment. The IVY network study found a VE against hospitalization of 75% (50-87%). The VISION study found a VE against hospitalization of 80% (71-85%) and VE against ED visit of 77% (70-83%). The VETERANS study found a VE against hospitalization of 82% (69-89%) and VE against ED visit of 77% (71-82%). These studies assessed the VE against hospitalization and ED, while the RENOIR trial did not, and they found values of VE lower than those assumed in your CE study 84.6% (32.0-98.3). The mean value of VE against hospitalization from observational studies in this review is 78%, 8.5% lower than that assumed in your CE study and similar to that found in the JAMA study.

The CE study must use consistent data concerning the key data on the vaccine effectiveness against different outcomes assessed. This critical aspect is incorrect in the CE study because it assumes several undemonstrated equivalences between the outcome “medically attended RSV-associated LRTD” in in then RENOIR trial and the outcomes hospitalization, ED visit and O visit in your CE study. It is clear that the outcome “medically attended RSV-associated LRTD” in in then RENOIR trial cannot be directly assumed to be the same as hospitalization, ED visit and outpatient visits in your study. any outpatient or inpatient visit such as hospitalization, emergency department visit, urgent care visit, home health care services, primary care physician office visit, pulmonologist office visit, specialist office visit, other visit, or telehealth contact cannot be the same as hospitalization and ED visits. This assumption is a critical mistake because consistent values of VE against hospitalization and ED visits (and OV visits) obtained in the Suri review, JAMA study or in other studies could have been used in the CE study. 

Lines 107-412. The limitations of the study should mention that the CE results can favor the vaccine because the outcome from the RENOIR trial cannot be equivalent to hospitalization and ED visits, resulting in a potentially greater biased VE against hospitalization and ED visits. References 59-61 (Surie review) found values of VE against hospitalization and ED visits lower than those assumed in the CE study. 

Consequently, I recommend to do again the CE analysis using consistent values of VE against hospitalization and ED visits (and OV visits) obtained in the Suri review, JAMA study or in other studies.

Comments on the Quality of English Language

The English is correct

Author Response

Reviewer: 1

Reviewer(s)' Comments to Author:

Your responses to the two critical questions are not adequate. The following two questions require a consistent response: 1) Why the VE against “any outpatient or inpatient visit” in the RENOIR trial is equivalent to the VE against hospitalization and the VE against ED visit, if the RENOIR was not able to assess the VE against these outcomes? 2) Why the VE against hospitalization obtained in the JAMA study cannot be a better and consistent assumption for the VE in your CE study? As your responses to these two questions are not clear and consistent, the CE analysis can be potentially incorrect or biased. 

Consequently, the CE analysis must be done again using consistent values of VE against hospitalization and ED visits (and OV visits) obtained in the Suri review, JAMA study or in other studies.

You comment that “Vaccine efficacy against severe RSV-LRTIs like those requiring hospitalization or oxygen supplementation was a secondary endpoint, therefore not powered to estimate VE specifically for hospitalization alone. However, recent real-world effectiveness on hospitalization [1] is now emerging, and despite its limitations (severe comorbid population with 24% immunocompromised and median of 5 for Charlson Comoribidity Index) and the short follow-up period (median time from vaccination to illness of 84 days), the real-world effectiveness data are largely aligned with the RSV-LRTI VE observed in the RENOIR trial for medically-attended RSV-LRTI cases with ≥3 symptoms.”  supports my critical comment. The reference (1) in your response is the JAMA study where the VE against hospitalization was 75% (95% CI: 50-87%). In fact, the JAMA study shows that real-world data for VE against hospitalization can be different than the VE against medically-attended RSV-LRTI cases with ≥3 symptoms in the RENOIR trial.  The VE of 75% (95% CI: 50-87%) is 12.8% lower than the VE of 84.6% (95% CI: 32.0-98.3) in your CE study. Again, VE against “any outpatient or inpatient visit” in the RENOIR trial cannot be equivalent to the VE against hospitalization, unless this equivalence is demonstrated in a consistent way. This demonstartion has nor been presented in the Methods section.

The Surie review (reference 3 in your response) supports my critical comment. The IVY network study found a VE against hospitalization of 75% (50-87%). The VISION study found a VE against hospitalization of 80% (71-85%) and VE against ED visit of 77% (70-83%). The VETERANS study found a VE against hospitalization of 82% (69-89%) and VE against ED visit of 77% (71-82%). These studies assessed the VE against hospitalization and ED, while the RENOIR trial did not, and they found values of VE lower than those assumed in your CE study 84.6% (32.0-98.3). The mean value of VE against hospitalization from observational studies in this review is 78%, 8.5% lower than that assumed in your CE study and similar to that found in the JAMA study.

The CE study must use consistent data concerning the key data on the vaccine effectiveness against different outcomes assessed. This critical aspect is incorrect in the CE study because it assumes several undemonstrated equivalences between the outcome “medically attended RSV-associated LRTD” in in then RENOIR trial and the outcomes hospitalization, ED visit and O visit in your CE study. It is clear that the outcome “medically attended RSV-associated LRTD” in in then RENOIR trial cannot be directly assumed to be the same as hospitalization, ED visit and outpatient visits in your study. any outpatient or inpatient visit such as hospitalization, emergency department visit, urgent care visit, home health care services, primary care physician office visit, pulmonologist office visit, specialist office visit, other visit, or telehealth contact cannot be the same as hospitalization and ED visits. This assumption is a critical mistake because consistent values of VE against hospitalization and ED visits (and OV visits) obtained in the Suri review, JAMA study or in other studies could have been used in the CE study.

Reply: We appreciate your comments. We would like to note that we have conducted a sensitivity analysis, which demonstrates the robustness of our results. Moreover, the real-world data provided by Surie et al. (75% VE in the JAMA study) do not offer Abrysvo-specific estimates. In addition, we have acknowledged in the limitations section that due to a limited number of hospitalized RSV cases in the RENOIR study, the VE against medically-attended RSV-LRTI with ≥3 symptoms was used as a surrogate for VE against hospitalized RSV. While there may be concerns about a mismatch, recent early-use real-world effectiveness data against RSV hospitalization report a similar magnitude of protection as those observed in the RENOIR clinical trial. Recent Abrysvo-specific real-world effectiveness (91%) reported by Tartof et al. (Kaiser Permanente) suggests that efficacy data in the RENOIR clinical trial closely approximates its real-world effectiveness in preventing more severe outcomes, like hospitalizations.

Reviewer: 1

Lines 107-412. The limitations of the study should mention that the CE results can favor the vaccine because the outcome from the RENOIR trial cannot be equivalent to hospitalization and ED visits, resulting in a potentially greater biased VE against hospitalization and ED visits. References 59-61 (Surie review) found values of VE against hospitalization and ED visits lower than those assumed in the CE study.

Reply: Based on the data by the Tartof et al. (Kaiser Permanente), Abrysvo-specific real-world VE data (91% from Tartof et al.) suggests that the RENOIR trial may underestimate the effect of the vaccine, particularly against severe outcomes like hospitalization. Therefore, our CE analysis, based on the RENOIR trial data (with an 84.6% VE), might actually present a conservative view of the vaccine’s cost-effectiveness. The use of composite endpoints in RENOIR is acknowledged as a limitation, but the model’s assumptions are consistent with the available trial and real-world data. The Surie review reports VE data for other RSV vaccines or populations, which may not be directly applicable to Abrysvo.

Reviewer: 1

Consequently, I recommend to do again the CE analysis using consistent values of VE against hospitalization and ED visits (and OV visits) obtained in the Suri review, JAMA study or in other studies.

Reply: The effectiveness estimates reported by Surie et al are not specific to Abrysvo. Abrysvo-specific real-world data (Tartof et al.) supports a higher effectiveness against hospitalization than the efficacy data from RENOIR, suggesting that the CE analysis based on RENOIR’s VE may underestimate the vaccine's benefits. This provides a more conservative scenario rather than an overestimation. It is also important to note that the same model and data, were approved by the Greek Health Technology Assessment (HTA) and received a positive recommendation for Abrysvo. Moreover, as we reported the RENOIR trial data has been utilized in previous cost-effectiveness studies as a proxy for estimating vaccine efficacy against more severe outcomes, such as hospitalization. Hence, considered appropriate for these analyses which informed for instance the Canadian National Advisory Committee on Immunization and the US Advisory Committee on Immunization Practices (ACIP).

Round 3

Reviewer 1 Report

Comments and Suggestions for Authors

The article gas sufficient scientific merit, interest and relevance for its publication in the journal Vaccines.

The methods/rationale for determining the effectiveness of a vaccine or medical intervention in a CE study is a important and critical methodological aspect in a CE study. The lack of data on the VE against hospitalization in the RENOIR trial is a great challenge for assessing the vaccine CE.      

The quality of the article would be greater if a consistent method/rationale had been used for determining the key values of VE against hospitalization, ED visits and OV visits in the CE study.

However, the inclusion of a comment in the Methods section about the reasons for using the VE against "medically-attended RSV-LRTI with ≥3 symptoms" a surrogate for VE against hospitalized RSV, and the in inclusion of a paragraph on the study limitations in the Discussion section about the potential limitations derived from using the VE against "medically-attended RSV-LRTI with ≥3 symptoms" in the RENOIR trial as equivalent to the VE against hospitalization and ED visits can be considered sufficient for avoiding potential complains against this use.